# On the Expressiveness and Generalization of Hypergraph Neural Networks

**Zhezheng Luo**     **Jiayuan Mao**     **Joshua B. Tenenbaum**     **Leslie Pack Kaelbling**

Massachusetts Institute of Technology

{ezzluo,jiayuanm,jbt}@mit.edu  lpk@csail.mit.edu

## Abstract

This extended abstract describes a framework for analyzing the expressiveness, learning, and (structural) generalization of hypergraph neural networks (Hyper-GNNs). Specifically, we focus on how HyperGNNs can learn from finite datasets and generalize structurally to graph reasoning problems of arbitrary input sizes. Our first contribution is a fine-grained analysis of the expressiveness of Hyper-GNNs, that is, the set of functions that they can realize. Our result is a hierarchy of problems they can solve, defined in terms of various hyperparameters such as depths and edge arities. Next, we analyze the learning properties of these neural networks, especially focusing on how they can be trained on a finite set of small graphs and generalize to larger graphs, which we term structural generalization. Our theoretical results are further supported by the empirical results.

## 1   Introduction

Reasoning over graph-structured data is an important task in many applications, including molecule analysis, social network modeling, and knowledge graph reasoning [1–3]. While we have seen great success of various relational neural networks, such as Graph Neural Networks [GNNs; 4] and Neural Logical Machines [NLM; 5] in a variety of applications [6–8], we do not yet have a full understanding of how different design parameters, such as the depth of the neural network, affects the expressiveness of these models, or how effectively these models generalize from limited data.

This paper analyzes the *expressiveness* and *generalization* of relational neural networks applied to *hypergraphs*, which are graphs with edges connecting more than two nodes. Literature has shown that even when the inputs and outputs of models have only unary and binary relations, allowing intermediate hyperedge representations increases the expressiveness [9, 10]. In this paper, we further formally show the "if and only if" conditions for the expressive power with respect to the edge arity. That is, $k$-ary hyper-graph neural networks are sufficient and necessary for realizing FOC-$k$, a fragment of first-order logic with counting quantification which involves at most $k$ variables. This is a helpful result because now we can determine whether a specific hypergraph neural network can solve a problem by understanding what form of logic formula can represent the solution to this problem. Next, we formally described the relationship between expressiveness and non-constant-depth networks. We state a conjecture about the "depth hierarchy," and connect the potential proof of this conjecture to the distributed computing literature.

Furthermore, we prove, under certain assumptions, it is possible to train a hypergraph neural networks on a finite set of small graphs, and it will generalize to arbitrarily large graphs. This ability results from the weight-sharing nature of hypergraph neural networks. We hope our work can serve as a foundation for designing hypergraph neural networks: to solve a specific problem, what arity do you need? What depth do you need? Will my model have structural generalization (i.e., to larger graphs)? Our theoretical results are further supported by experiments, for empirical demonstrations.

Z. Luo et al., On the Expressiveness and Generalization of Hypergraph Neural Networks (Extended Abstract).
Presented at the First Learning on Graphs Conference (LoG 2022), Virtual Event, December 9–12, 2022.

## 2 Hypergraph Reasoning Problems and Hypergraph Neural Networks

A *hypergraph representation* $G$ is a tuple $(V, X)$, where $V$ is a set of entities (nodes), and $X$ is a set of *hypergraph representation functions*. Specifically, $X = \{X_0, X_1, X_2, \cdots, X_k\}$, where $X_j : (v_1, v_2, \cdots, v_j) \rightarrow \mathcal{S}$ is a function mapping every tuple of $j$ nodes to a value. We call $j$ the *arity* of the hyperedge and $k$ is the max arity of input hyperedges. The range $\mathcal{S}$ can be any set of discrete labels that describes relation type, or a scalar number (e.g., the length of an edge), or a vector. We will use the arity 0 representation $X_0(\emptyset) \rightarrow \mathcal{S}$ to represent any global properties of the graph.

A *graph reasoning function* $f$ is a mapping from a hypergraph representation $G = (V, X)$ to another hyperedge representation function $Y$ on $V$. As concrete examples, asking whether a graph is fully connected is a graph classification problem, where the output $Y = \{Y_0\}$ and $Y_0(\emptyset) \rightarrow \mathcal{S}' = \{0, 1\}$ is a global label; finding the set of disconnected subgraphs of size $k$ is a $k$-ary hyperedge classification problem, where the output $Y = \{Y_k\}$ is a label for each $k$-ary hyperedges.

There are two main motivations and constructions of a neural network applied to graph reasoning problems: message-passing-based and first-order-logic-inspired. Both approaches construct the computation graph layer by layer. The input is the features of nodes and hyperedges, while the output is the per-node or per-edge prediction of desired properties, depending on the task.

In a nutshell, within each layer, *message-passing-based* hypergraph neural networks, Higher-Order GNNs [11], perform message passing between each hyperedge and its neighbours. Specifically, we say the j-th neighbour set of a hyperedge $u = (x_1, x_2, \cdots, x_i)$ of arity $i$ is $N_j(u) = \{(x_1, x_2, \cdots, x_{j-1}, r, x_{j+1}, \cdots, x_i)\}$, where $r \in V$. Then, the all neighbours of node $u$ is the union of all $N_j$'s, where $j = 1, 2, \cdots, i$.

On the other hand, first-order-logic-inspired hypergraph neural networks consider building neural networks that can emulate first logic formulas. Neural Logic Machines [NLM; 5] are defined in terms of a set of input hyperedges; each hyperedge of arity $k$ is represented by a vector of (possibly real) values obtained by applying all of the k-ary predicates in the domain to the tuple of vertices it connects. Each layer in an NLM learns to apply a linear transformation with nonlinear activation and quantification operators (analogous to the for all $\forall$ and exists $\exists$ quantifiers in first-order logic), on these values. It is easy to prove, by construction, that given a sufficient number of layers and maximum arity, NLMs can learn to realize any first-order-logic formula. For readers who are not familiar with HO-GNNs [11] and NLMs [5], we include a mathematical summary of their computation graph in Appendix A. Our analysis starts from the following theorem.

**Theorem 2.1.** HO-GNNs [11] are equivalent to NLMs in terms of expressiveness. Specifically, a $B$-ary HO-GNN is equivalent to an NLM applied to $B + 1$-ary hyperedges. Proofs are in Appendix A.3.

Given Theorem 2.1, we can focus on just one single type of hypergraph neural network. Specifically, we will focus on Neural Logic Machines [NLM; 5] because its architecture naturally aligns with first-order logic formula structures, which will aid some of our analysis. An NLM is characterized by hyperparameters $D$ (depth), and $B$ maximum arity. We are going to assume that $B$ is a constant, but $D$ can be dependent on the size of the input graph. We will use NLM[$D$, $B$] to denote an NLM family with depth $D$ and max arity $B$. Other parameters such as the width of neural networks affects the precise details of what functions can be realized, as it does in a regular neural network, but does not affect the analyses in this extended abstract. Furthermore, we will be focusing on neural networks with bounded precision, and briefly discuss how our results generalize to unbounded precision cases.

## 3 Expressiveness of Relational Neural Networks

We start from a formal definition of hypergraph neural network expressiveness.

**Definition 3.1** (Expressiveness). We say a model family $\mathcal{M}_1$ is *at least expressive as* $\mathcal{M}_2$, written as $\mathcal{M}_1 \succcurlyeq \mathcal{M}_2$, if for all $M_2 \in \mathcal{M}_2$, there exists $M_1 \in \mathcal{M}_1$ such that $M_1$ can realize $M_2$. A model family $\mathcal{M}_1$ is *more expressive than* $\mathcal{M}_2$, written as $\mathcal{M}_1 \succ \mathcal{M}_2$, if $\mathcal{M}_1 \succcurlyeq \mathcal{M}_2$ and $\exists M_1 \in \mathcal{M}_1$, $\forall M_2 \in \mathcal{M}_2$, $M_2$ can not realize $M_1$.

**Arity Hierarchy** We first aim to quantify how the maximum arity $B$ of the network's representation affects its expressiveness and find that, in short, even if the inputs and outputs of neural networks are of low arity, the higher the maximum arity for intermediate layers, the more expressive the NLM is.

**Corollary 3.1** (Arity Hierarchy). For any maximum arity $B$, there exists a depth $D^*$ such that: $\forall D \geq D^*$, NLM[$D$, $B + 1$] is more expressive than NLM[$D$, $B$]. This theorem applies to both

fixed-precision and unbounded-precision networks. Here, by fixed-precision, we mean that the results of intermediate layers (tensors) are constant-sized (e.g., $W$ bits per entry). Practical GNNs are all fixed-precision because real number types in modern computers have finite precision.

*Proof sketch:* Our proof slightly extends the proof of Morris et al. [11]. First, the set of graphs distinguishable by NLM[$D, B$] is bounded by graphs distinguishable by a $D$-round order-$B$ Weisfeiler-Leman test [12]. If models in NLM[$D, B$] cannot generate different outputs for two distinct hypergraphs $G_1$ and $G_2$, but there exists $M \in$ NLM[$D, B + 1$] that *can* generate different outputs for $G_1$ and $G_2$, then we can construct a graph classification function $f$ that NLM[$D, B + 1$] (with some fixed precision) can realize but NLM[$D, B$] (even with unbounded precision) cannot.* The full proof is described in Appendix B.1.

It is also important to quantify the minimum arity for realizing certain graph reasoning functions.
**Corollary 3.2** (FOL realization bounds). Let FOC$_B$ denote a fragment of first order logic with at most $B$ variables, extended with counting quantifiers of the form $\exists^{\geq n} \phi$, which state that there are at least $n$ nodes satisfying formula $\phi$ [13].

- (Upper Bound) Any function $f$ in FOC$_B$ can be realized by NLM[$D, B$] for some $D$.
- (Lower Bound) There exists a function $f \in$ FOC$_B$ such that for all $D$, $f$ cannot be realized by NLM[$D, B - 1$].

*Proof:* The upper bound part of the claim has been proved by Barceló et al. [14] for $B = 2$. The results generalize easily to arbitrary $B$ because the counting quantifiers can be realized by sum aggregation. The lower bound part can be proved by applying Section 5 of [13], in which they show that FOC$_B$ is equivalent to a $(B - 1)$-dimensional WL test in distinguishing non-isomorphic graphs. Given that NLM[$D, B - 1$] is equivalent to the $(B - 2)$-dimensional WL test of graph isomorphism, there must be an FOL$_B$ formula that distinguishes two non-isomorphic graphs that NLM[$D, B - 1$] cannot. Hence, FOL$_B$ cannot be realized by NLM[$\cdot, B - 1$].

**Depth Hierarchy** We now study the dependence of the expressiveness of NLMs on depth $D$. Neural networks are generally defined to have a fixed depth, but allowing them to have a depth that is dependent on the number of nodes $n = |V|$ in the graph, in many cases, can substantially increase their expressive power [15, see also Theorem 3.4 and Appendix B for examples]. In the following, we define a *depth hierarchy* by analogy to the *time hierarchy* in computational complexity theory [16], and we extend our notation to let NLM[$O(f(n)), B$] denote the class of adaptive-depth NLMs in which the growth-rate of depth $D$ is bounded by $O(f(n))$.
**Conjecture 3.3** (Depth hierarchy). For any maximum arity $B$, for any two functions $f$ and $g$, if $g(n) = o(f(n)/\log n)$, that is, $f$ grows logarithmically more quickly than $g$, then fixed-precision NLM[$O(f(n)), B$] is more expressive than fixed-precision NLM[$O(g(n)), B$].

There is a closely related result for the *congested clique* model in distributed computing, where [17] proved that CLIQUE($g(n)$) $\subsetneq$ CLIQUE($f(n)$) if $g(n) = o(f(n))$. This result does not have the $\log n$ gap because the congested clique model allows $\log n$ bits to transmit between nodes at each iteration, while fixed-precision NLM allows only a constant number of bits. The reason why the result on congested clique can not be applied to fixed-precision NLMs is that congested clique assumes unbounded precision representation for each individual node.

However, Conjecture 3.3 is not true for NLMs with unbounded precision, because there is an upper bound depth $O(n^{B-1})$ for a model's expressiveness power (see appendix B.2 for a formal statement and the proof). That is, an unbounded-precision NLM can not achieve stronger expressiveness by increasing its depth beyond $O(n^{B-1})$.

It is important to point out that, to realize a specific graph reasoning function, NLMs with different maximum arity $B$ may require different depth $D$. Fürer [18] provides a general construction for problems that higher-dimensional NLMs can solve in asymptotically smaller depth than lower-dimensional NLMs. In the following we give a concrete example for computing *S-T Connectivity-$k$*, which asks whether there is a path of nodes from $S$ and $T$ in a graph, with length $\leq k$.
**Theorem 3.4** (S-T Connectivity-$k$ with Different Max Arity). For any function $f(k)$, if $f(k) = o(k)$, NLM[$O(f(k)), 2$] cannot realize S-T Connectivity-$k$. That is, S-T Connectivity-$k$ requires depth at least $O(k)$ for a relational neural network with an maximum arity of $B = 2$. However, S-T Connectivity-$k$ *can* be realized by NLM[$O(\log k), 3$].

---

*Note that the arity hierarchy is applied to fixed-precision and unbounded-precision separately. For example, NLM[$D, B$] with unbounded precision is incomparable with NLM[$D, B + 1$] with fixed precision.

*Proof sketch.* For any integer $k$, we can construct a graph with two chains of length $k$, so that if we mark two of the four ends as $S$ or $T$, any NLM$[k-1, 2]$ cannot tell whether $S$ and $T$ are on the same chain. The full proof is described in Appendix B.3.

There are many important graph reasoning tasks that do not have known depth lower bounds, including all-pair connectivity and shortest distance [19, 20]. In Appendix B.3, we discuss the concrete complexity bounds for a series of graph reasoning problems.

## 4 Learning and Generalization in Relational Neural Networks

Given our understanding of what functions can be realized by NLMs, we move on to the problems of learning them: Can we effectively learn a NLMs to solve a desired task given a sufficient number of input-output examples? In this paper, we show that applying *enumerative training* with examples up to some fixed graph size can ensure that the trained neural network will generalize to all graphs *larger* than those appearing in the training set.

A critical determinant of the generalization ability for NLMs is the aggregation function. Specifically, Xu et al. [21] have shown that using *sum* as the aggregation function provides maximum expressiveness for graph neural networks. However, sum aggregation cannot be implemented in fixed-precision models, because as the graph size $n$ increases, the range of the sum aggregation also increases.

**Definition 4.1** (Fixed-precision aggregation function). An aggregation function is *fixed precision* if it maps from any finite *set* of inputs with values drawn from *finite domains* to a *fixed finite* set of possible output values; that is, the cardinality of the range of the function cannot grow with the number of elements in the input set. Two useful fixed-precision aggregation functions are *max*, which computes the dimension-wise maximum over the set of input values, and *fixed-precision mean*, which approximates the dimension-wise mean to a fixed decimal place.

In order to focus on structural generalization in this section, we consider an *enumerative* training paradigm. When the input hypergraph representation domain $\mathcal{S}$ is a finite set, we can enumerate the set $\mathcal{G}_{\leq N}$ of all possible input hypergraph representations of size bounded by $N$. We first enumerate all graph sizes $n \leq N$; for each $n$, we enumerate all possible values assigned to the hyperedges in the input. Given training size $N$, we enumerate all inputs in $\mathcal{G}_{\leq N}$, associate with each one the corresponding ground-truth output representation, and train the model with these input-output pairs.

This has much stronger data requirements than the standard sampling-based training mechanisms in machine learning. In practice, this can be approximated well when the input domain $\mathcal{S}$ is small and the input data distribution is approximately uniformly distributed. The enumerative learning setting is studied by the *language identification in the limit* community [22], in which it is called *complete presentation*. This is an interesting learning setting because even if the domain for each individual hyperedge representation is finite, as the graph size can go arbitrarily large, the number of possible inputs is enumerable but unbounded.

**Theorem 4.1** (Fixed-precision generalization under complete presentation). For any hypergraph reasoning function $f$, if it can be realized by a fixed-precision relational neural network model $\mathcal{M}$, then there exists an integer $N$, such that if we train the model with complete presentation on all input hypergraph representations with size smaller than $N$, $\mathcal{G}_{\leq N}$, then for all $M \in \mathcal{M}$,

$$\sum_{G \in \mathcal{G}_{\leq N}} 1[M(G) \neq f(G)] = 0 \implies \forall G \in \mathcal{G}_{\infty} : M(G) = f(G).$$

That is, as long as $M$ fits all training examples, it will generalize to all possible hypergraphs in $\mathcal{G}_{\infty}$.

*Proof.* The key observation is that for any fixed vector representation length $W$, there are only a finite number of distinctive models in a fixed-precision NLM family, *independent of the graph size $n$*. Let $W_b$ be the number of bits in each intermediate representation of a fixed-precision NLM. There are at most $(2^{W_b})^{2^{W_b}}$ different mappings from inputs to outputs. Hence, if $N$ is sufficiently large to enumerate all input hypergraphs, we can always identify the correct model in the hypothesis space.

Our results are related to the *algorithmic alignment* approach [23, 24]. In contrast to their Probably Approximately Correct (PAC) Learning bounds for sample efficiency, our expressiveness results directly quantifies whether a hypergraph neural network can be trained to realize a specific function.

## 5   Related Work

Solving problems on graphs of arbitrary size is studied in many fields. NLMs can be viewed as circuit families with constrained architecture. In distributed computation, the congested clique model can be viewed as 2-arity NLMs, where nodes have identities as extra information. Common graph problems including sub-structure detection[25, 26] and connectivity[19] are studied for lower bounds in terms of depth, width and communication. This has been connected to GNNs for deriving expressiveness bounds [27].

Studies have been conducted on the expressiveness of GNNs and their variants. Xu et al. [21] provide an illuminating characterization of GNN expressiveness in terms of the WL graph isomorphism test. Azizian and Lelarge [9] analyze the expressiveness of higher-order Folklore GNNs by connecting them with high-dimensional WL-tests. We have the similar results in the arity hierarchy. Barceló et al. [14] reviewed GNNs from the logical perspective and rigorously refined their logical expressiveness with respect to fragments of first-order logic. Dong et al. [5] proposed Neural Logical Machines (NLMs) to reason about higher-order relations, and showed that increasing order inreases expressiveness. It is also possible to gain expressiveness using unbounded computation time, as shown by the work of Dehghani et al. [15] on dynamic halting in transformers.

It is interesting that GNNs may generalize to larger graphs. Xu et al. [23, 24] have studied the notion of *algorithmic alignment* to quantify such structural generalization. Dong et al. [5] provided empirical results showing that NLMs generalize to much larger graphs on certain tasks. Buffelli et al. [28] introduced a regularization technique to improve GNNs' generalization to larger graphs and demonstrated its effectiveness empirically. In Xu et al. [23], they analyzed and compared the sample complexity of Graph Neural Networks. This is different from our notion of expressiveness for realizing functions. In Xu et al. [24], they showed emperically on some problems (e.g., Max-Degree, Shortest Path, and n-body problem) that algorithm alignment helps GNNs to extrapolate, and theoretically proved the improvement by algorithm alignment on the Max-Degree problem. In this extended abstract, instead of focusing on computing specific graph problems, we analyzed how GNNs can extrapolate to larger graphs in a general case, based on the assumption of fixed precision computation.

## 6   Conclusion

In this extended abstract, we have shown the substantial increase of expressive power due to higher-arity relations and increasing depth, and have characterized very powerful structural generalization from training on small graphs to performance on larger ones. All theoretical results are further supported by the empirical results, discussed in Appendix C. Although many questions remain open about the overall generalization capacity of these models in continuous and noisy domains, we believe this work has shed some light on their utility and potential for application in a variety of problems.

**Acknowledgement.**   We thank anonymous reviewers for their comments. This work is in part supported by ONR MURI N00014-16-1-2007, the Center for Brain, Minds, and Machines (CBMM, funded by NSF STC award CCF-1231216), NSF grant 2214177, AFOSR grant FA9550-22-1-0249, ONR grant N00014-18-1-2847, the MIT Quest for Intelligence, MIT–IBM Watson Lab. Any opinions, findings, and conclusions or recommendations expressed in this material are those of the authors and do not necessarily reflect the views of our sponsors.

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

# Appendix

The appendix is organized as the following. In Appendix A, we provide a formalization of two types of hypergraph neural networks discussed in the main paper, and proved their equivalence. In Appendix B, we prove the theorems for the arity hierarchy and provide concrete examples for expressiveness analyses. Finally, in Appendix C, we include additional experiment results to empirically illustrate the application of theorems discussed in the paper.

## A    Hypergraph Neural Networks

We now introduce two important hypergraph neural network implementations that can be trained to solve graph reasoning problems: Higher-order Graph Neural Networks [HO-GNN; 11] and Neural Logic Machines [NLM; 5]. The are equivalent to each other in terms of expressiveness. Showing this equivalence allows us to focus the rest of the paper on analyzing a single model type, with the understanding that the conclusions generalize to a broader class of hypergraph neural networks.

### A.1    Higher-order Graph Neural Networks

Higher-order Graph Neural Networks [HO-GNNs; 11] are Graph Neural Networks (GNNs) that apply to hypergraphs. A GNN is usually defined based on two message passing operations.

- Edge update: the feature of each edge is updated by features of its ends.
- Note update: the feature of each node is updated by features of all edges adjacent to it.

However, computing only node-wise and edge-wise features does not handle higher-order relations, such as triangles in the graph. In order to obtain more expressive power, GNNs have be extend to hypergraphs of higher arity [11]. Specifically, HO-GNNs on $B$-ary hypergraph maintains features for all $B$-tuple of nodes, and the neighborhood is extended to $B$-tuples accordingly: the feature of tuple $(v_1, v_2, \cdots, v_B)$ is updated by the $|V|$ element multiset (contain $|V|$ elements for each $u \in V$) of $B$-tuples of features

$$(H_i[u, v_2, \cdots, v_B], H_{i-1}[v_1, u, v_2, \cdots, v_B], \cdots H_{i-1}[v_1, \cdots, v_{B-1}, u]) \tag{A.1}$$

where $H_{i-1}[\boldsymbol{v}]$ is the feature of tuple $\boldsymbol{v}$ from the previous iteration.

We now introduce the formal definition of the high-dimensional message passing. We denote $\boldsymbol{v}$ as a $B$-tuple of nodes $(v_1, v_2, \cdots, v_B)$, and generalize the neighborhood to a higher dimension by defining the neighborhood of $\boldsymbol{v}$ as all node tuples that differ from $\boldsymbol{v}$ at one position.

$$\text{Neighbors}(\boldsymbol{v}, u) = ((u, v_2, \cdots, v_B), (v_1, u, v_3, \cdots, v_B), \cdots, (v_1, \cdots, v_{B-1}, u)) \tag{A.2}$$
$$N(\boldsymbol{v}) = \{\text{Neighbors}(\boldsymbol{v}, u) | u \in V\} \tag{A.3}$$

Then message passing scheme naturally generalizes to high-dimensional features using the high-dimensional neighborhood.

$$\text{Received}_i[\boldsymbol{v}] = \sum_u \left( \text{NN}_1 \left( H_{i-1}[\boldsymbol{v}]; \text{CONCAT}_{\boldsymbol{v}' \in \text{neighbors}(\boldsymbol{v}, u)} H_{i-1}[\boldsymbol{v}'] \right) \right) \tag{A.4}$$

### A.2    Neural Logic Machines

A NLM is a multi-layer neural network that operates on hypergraph representations, in which the hypergraph representation functions are represented as tensors. The input is a hypergraph representation $(V, X)$. There are then several computational layers, each of which produces a hypergraph representation with nodes $V$ and a new set of representation functions. Specifically, a $B$-ary NLM produces hypergraph representation functions with arities from 0 up to a maximum hyperedge arity of $B$. We let $T_{i,j}$ denote the tensor representation for the output at layer $i$ and arity $j$. Each entry in the tensor is a mapping from a set of node indices $(v_1, v_2, \cdots, v_j)$ to a vector in a latent space $\mathbb{R}^W$. Thus, $T_{i,j}$ is a tensor of $j+1$ dimensions, with the first $j$ dimensions corresponding to $j$-tuple of nodes, and the last feature dimension. For convenience, we write $h_{0,\cdot}$ for the input hypergraph representation and $h_{D,\cdot}$ for the output of the NLM.

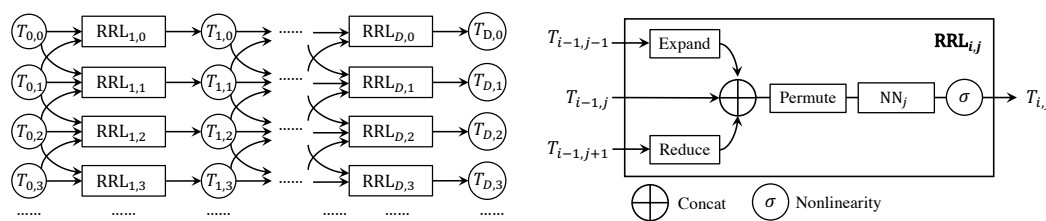

**(a)** The overall computation graph of a NLM.

**(b)** The computation graph of a single NLM block.

**Figure 1:** The overall architecture of our Neural Logic Machines (NLMs). It follows the computation graph of NLM [5] and can be applied to hypergraphs.

Fig. 1a shows the overall architecture of NLMs. It has $D \times B$ computation blocks, namely relational reasoning layers (RRLs). Each block $\text{RRL}_{i,j}$, illustrated in Fig. 1b, takes the output from neighboring arities in the previous layer, $T_{i-1,j-1}$, $T_{i-1,j}$ and $T_{i-1,j+1}$, and produces $T_{i,j}$. Below we show the computation of each primitive operation in an RRL.

The *expand* operation takes tensor $T_{i-1,j-1}$ (arity $j-1$) and produces a new tensor $T^E_{i-1,j-1}$ of arity $j$. The *reduce* operation takes tensor $T_{i-1,j+1}$ (arity $j+1$) and produces a new tensor $T^R_{i-1,j+1}$ of arity $j+1$. Mathematically,

$$
\begin{aligned}
T^E_{i-1,j-1}[v_1, v_2, \cdots, v_j] &= T_{i-1,j-1}[v_1, v_2, \cdots, v_{j-1}]; \\
T^R_{i-1,j+1}[v_1, v_2, \cdots, v_j] &= \text{Agg}_{v_{j+1}} \left\{ T_{i-1,j+1}[v_1, v_2, \cdots, v_j, v_{j+1}] \right\}.
\end{aligned}
$$

Here, Agg is called the aggregation function of a NLM. For example, a sum aggregation function takes the summation along the dimension $j+1$ of the tensor, and a max aggregation function takes the max along that dimension.

The *concat* (concatenate) operation $\bigoplus$ is applied at the "vector representation" dimension. The *permute* operation generates a new tensor of the same arity, but it fuses the representations of hyperedges that share the same set of entities but in different order, such as $(v_1, v_2)$ and $(v_2, v_1)$. Mathematically, for tensor $X$ of arity $j$, if $Y = \text{permute}(X)$ then

$$
Y[v_1, v_2, \cdots, v_j] = \underset{\sigma \in S_j}{\text{Concat}} \left\{ X[v_{\sigma_1}, v_{\sigma_2}, \cdots, v_{\sigma_j}] \right\},
$$

where $\sigma \in S_j$ iterates over all permuations of $\{1, 2, \cdots j\}$. $\text{NN}_j$ is a multi-layer perceptron (MLP) applied to each entry in the tensor produced after permutation, with nonlinearity $\sigma$ (e.g., ReLU).

It is important to note that we intentionally name the MLPs $\text{NN}_j$ instead of $\text{NN}_{i,j}$. In generalized relational neural networks, for a given arity $j$, all MLPs across all layers $i$ are shared. It is straightforward to see that this "weight-shared" model can realize a "non-weight-shared" NLM that uses different weights for MLPs at different layers when the number of layers is a constant. With a sufficiently large length of the representation vector, we can simulate the computation of applying different transformations by constructing block matrix weights. (A more formal proof is in Appendix A) The advantage of this weight sharing is that the network can be easily extended to a "recurrent" model. For example, we can apply the NLM for a number of layers that is a function of $n$, where $n$ is the the number of nodes in the input graph. Thus, we will use the term *layers* and *iterations* interchangeably.

Handling high-arity features and using deeper models usually increase the computational cost. In appendix A.5, we show that the time and space complexity of NLM $[D, B]$ is $O(Dn^B)$.

Note that even when hyperparameters such as the maximum arity and the number of iterations are fixed, a NLM is still a *model family* $\mathcal{M}$: the weights for MLPs will be trained on some data. Furthermore, each model $M \in \mathcal{M}$ is a NLM with a specific set of MLP weights.

### A.3 Expressiveness Equivalence of Relational Neural Networks

Since we are going to study both constant-depth and adaptive-depth graph neural networks, we first prove the following lemma (for general multi-layer neural networks), which helps us simplify the analysis.

**Lemma A.1.** A neural network with representation width $W$ that has $D$ different layers $\mathrm{NN}_1, \cdots, \mathrm{NN}_D$ can be realized by a neural network that applies a single layer $\mathrm{NN}'$ for $D$ iterations with width $(D+1)(W+1)$.

*Proof.* The representation for $\mathrm{NN}'$ can be partitioned into $D+1$ segments each of length $W+1$. Each segment consist of a "flag" element and a $W$-element representation, which are all $0$ initially, except for the first segment, where the flag is set to $1$, and the representation is the input.

$\mathrm{NN}'$ has the weights for all $\mathrm{NN}_1, \cdots, \mathrm{NN}_D$, where weights $\mathrm{NN}_i$ are used to compute the representation in segment $i+1$ from the representation in segment $i$. Additionally, at each iteration, segment $i+1$ can only be computed if the flag in segment $i$ is $1$, in which case the flag of segment $i+1$ is set to $1$. Clearly, after $D$ iterations, the output of $\mathrm{NN}_k$ should be the representation in segment $D+1$. □

Due to Lemma A.1, we consider the neural networks that recurrently apply the same layer because a) they are as expressive as those using layers of different weights, b) it is easier to analyze a single neural network layer than $D$ layers, and c) they naturally generalize to neural networks that runs for adaptive number of iterations (e.g. GNNs that run $O(\log n)$ iterations where $n$ is the size of the input graph).

We first describe a framework for quantifying if two hypergraph neural network models are equally expressive on regression tasks (which is more general than classification problems). The framework view the expressiveness from the perspective of computation. Specifically, we will prove the expressiveness equivalence between models by showing that their computation can be aligned.

In complexity, we usually show a problem is at least as hard as the other one by showing a reduction from the other problem to the problem. Similarly, on the expressiveness of NLMs, we can construct reduction from model family $\mathcal{A}$ to model family $\mathcal{B}$ to show that $\mathcal{B}$ can realize all computation that $\mathcal{A}$ does, or even more. Formally, we have the following definition.

**Definition A.1** (Expressiveness reduction). For two model families $\mathcal{A}$ and $\mathcal{B}$, we say $\mathcal{A}$ can be **reduced** to $\mathcal{B}$ if and only if there is a function $r : \mathcal{A} \to \mathcal{B}$ such that for each model instance $A \in \mathcal{A}$, $r(A) \in \mathcal{B}$ and $A$ have the same outputs on all inputs. In this case, we say $\mathcal{B}$ is at least as expressive as $\mathcal{A}$.

**Definition A.2** (Expressiveness equivalence). For two model families $\mathcal{A}$ and $\mathcal{B}$, if $\mathcal{A}$ and $\mathcal{B}$ can be reduced to each other, then $\mathcal{A}$ and $\mathcal{B}$ are equally expressive. Note that this definition of expressiveness equivalence generalizes to both classification and regression tasks.

**Equivalence between HO-GNNs and NLMs.** We will prove the equivalence between HO-GNNs and NLMs by making reductions in both directions.

**Lemma A.2.** A $B$-ary HO-GNN with depth $D$ can be realized by a NLM with maximum arity $B+1$ and depth $2D$.

*Proof.* We prove lemma A.2 by showing that one layer of GNNs on $B$-ary hypergraphs can be realized by two NLM with maximum arity $B+1$.

Firstly, a GNN layer maintain features of $B$-tuples, which are stored in correspondingly in an NLM layer at dimension $B$. Then we will realize the message passing scheme using the NLM features of dimension $B$ and $B+1$ in two steps.

Recall the message passing scheme generalized to high dimensions (to distinguish, we use $H$ for HO-GNN features and $T$ for NLM features.)

$$\mathrm{Received}_i(\boldsymbol{v}) = \sum_u \left( \mathrm{NN}_1 \left( H_{i-1,B}[\boldsymbol{v}]; \mathrm{CONCAT}_{\boldsymbol{v}' \in \mathrm{neighbors}(\boldsymbol{v},u)} H_{i-1}[\boldsymbol{v}'] \right) \right) \tag{A.5}$$

At the first step, the Expand operation first raise the dimension to $B + 1$ by expanding a non-related variable $u$ to the end, and the Permute operation can then swap $u$ with each of the elements (or no swap). Particularly, $T_{i,B}[v_1, v_2, \cdots, v_B]$ will be expand to

$$T_{i+1,B+1}[u, v_2, v_3, \cdots, v_B, v_1], T_{i+1,B+1}[v_1, u, v_3, \cdots, v_B, v_2], \cdots,$$
$$T_{i+1,B+1}[v_1, v_2, \cdots, v_{B-1}, u, v_B], \text{and } T_{i+1,B+1}[v_1, v_2, \cdots, v_{B-1}, v_B, u]$$

Hence, $T_{i+1,B+1}[v_1, v_2, v_3, \cdots, v_B, u]$ receives the features from

$$T_{i,B}[v_1, v_2, \cdots, v_B], T_{i,B}[u, v_2, v_3, \cdots, v_B], T_{i,B}[v_1, u, v_3, \cdots, v_B], \cdots, T_{i,B}[v_1, v_2, \cdots, v_{B-1}, u]$$

These features matches the input of $\mathrm{NN}_1$ in equation A.5, and in this layer $\mathrm{NN}_1$ can be applied to compute things inside the summation.

Then at the second step, the last element is reduced to get what tuple $v$ should receive, so $v$ can be updated. Since each HO-GNN layer can be realized by such two NLM layers, each $B$-ary HO-GNN with depth $D$ can be realized by a NLM of maximum arity $(B + 1)$ and depth $2D$. $\qquad \square$

To complete the proof we need to find a reduction from NLMs of maximum arity $B + 1$ to $B$-ary HO-GNNs. The key observation here is that the features of $(B + 1)$-tuples in NLMs can only be expanded from sub-tuples, and the expansion and reduction involving $(B+1)$-tuples can be simulated by the message passing process.

**Lemma A.3.** The features of $(B + 1)$-tuples feature $T_{i,B+1}[v_1, v_2, \cdots, v_{B+1}]$ can be computed from the following tuples

$$(T_{i,B}[v_2, v_3, \cdots, v_{B+1}], T_{i,B}[v_1, v_3, \cdots, v_{B+1}], \cdots, T_{i,B}[v_1, v_2, \cdots, v_B]).$$

*Proof.* Lemma A.3 is true because $(B + 1)$-dimensional representations can either be computed from themselves at the previous iteration, or expanded from $B$-dimensional representations. Since representations at all previous iterations $j < i$ can be contained in $T_{i,B}$, it is sufficient to compute $T_{i,B+1}[v_1, v_2, \cdots, v_{B+1}]$ from all its $B$-ary sub-tuples. $\qquad \square$

Then let's construct the HO-GNN for given NLM to show the existence of the reduction.

**Lemma A.4.** A NLM of maximum arity $B + 1$ and depth $D$ can be realized by a $B$-ary HO-GNN with no more than $D$ iterations.

*Proof.* We can realize the Expand and Reduce operation with only the $B$-dimensional features using the broadcast message passing scheme. Note that Expand and Reduce between $B$-dimensional features and $(B + 1)$-dimensional features in the NLM is a special case where claim A.3 is applied.

Let's start with Expand and Reduce operations between features of dimension $B$ or lower. For the $b$-dimensional feature in the NLM, we keep $n^{\underline{b}} n^{B-b}$[†] copies of it and store them the representation of every $B$-tuple who has a sub-tuple[‡] that is a permutation of the $b$-tuple. That is, for each $B$-tuple in the $B$-ary HO-GNN, for its every sub-tuple of length $b$, we store $b!$ representations corresponding to every permutation of the $b$-tuple in the NLM. Keeping representation for all sub-tuple permutations make it possible to realize the Permute operation. Also, it is easy to notice that Expand operation is realized already, as all features with dimension lower than $B$ are naturally expanded to $B$ dimension by filling in all possible combinations of the rest elements. Finally, the Reduce operation can be realized using a broadcast casting message passing on certain position of the tuple.

Now let's move to the special case – the Expand and Reduce operation between features of dimensions $B$ and $B + 1$. Claim A.3 suggests how the $(B + 1)$-dimensional features are stored in $B$-dimensional representations in GNNs, and we now show how the Reduce can be realized by message passing.

We first bring in claim A.3 to the HO-GNN message passing, where we have $\mathrm{Received}_i[v]$ to be

---

[†] $n^{\underline{k}} = n \times (n - 1) \times \cdots \times (n - k + 1)$.

[‡] The sub-tuple does not have to be consecutive, but instead can be a any subset of the tuple that keeps the element order.

$$\sum_u \left( \text{NN}_1 \left( T_{i-1,B}[v_2, v_3, \cdots, v_B, u], T_{i-1,B}[v_1, v_3, \cdots, v_B, u], \cdots, T_{(i-1),B}[v_1, v_2, \cdots, v_B] \right) \right)$$

Note that the last term $T_{i-1,B}[v_1, v_2, \cdots, v_B]$ is contained in $H_{i-1}(v)$ in equation A.5, and other terms are contained in $H_{i-1}(v')$ for $v' \in \text{neighbors}(\boldsymbol{v}, u)$. Hence, equation A.5 is sufficient to simulate the Reduce operation. □

**Theorem A.5.** $B$-ary HO-GNNs are equally expressive as NLMs with maximum arity $B + 1$.

*Proof.* This is a direct conclusion by combining Lemma A.2 and Lemma A.4. □

### A.4 Expressiveness of hypergraph convolution and attention

There exist other variants of hypergraph neural networks. In particular, hypergraph convolution[29–31], attention[32] and message passing[33] focus on updating node features instead of tuple features through hyperedges . These approaches can be viewed as instances of hypergraph neural networks, and they have smaller time complexity because they do not model all high-arity tuples. However, they are less expressive than the standard hypergraph neural networks with equal max arity.

These approaches can be formulated to two steps at each iteration. At the first step, each hyperedge is updated by the features of nodes it connects.

$$h_{i,e} = \text{AGG}_{v \in e} f_{i-1,v} \tag{A.6}$$

At the second step, each node is updated by the features of hyperedges connecting it.

$$f_{i,v} = \text{AGG}_{v \in e} h_{i,e} \tag{A.7}$$

where $f_{i,v}$ is the feature of node $v$ at iteration $i$, and $h_{i,v}$ is the aggregated message passing through hyperedge $e$ at iteration $i + 1$.

It is not hard to see that A.6 can be realized by $B$ iterations of NLM layers with Expand operations where $B$ is the max arity of hyperedges. This can be done by expanding each node feature to every high arity features that contain the node, and aggregate them at the tuple corresponding to each hyperedge. Then, A.7 can also be realized by $B$ iterations of NLM layers with Reduce operations, as the tuple feature will finally be reduced to a single node contained in the tuple.

This approach has lower complexity compared to the GNNs we study applied on hyperedges, because it only requires communication between nodes and hyperedges connecting to them, which takes $O(|V| \cdot |E|)$ time at each iteration. Compared to them, NLMs takes $O(|V|^B)$ time because NLMs keep features of every tuple with max arity $B$, and allow communication from tuples to tuples instead of between tuples and single nodes. An example is provided below that this approach can not solve while NLMs can.

Consider a graph with 6 nodes and 6 edges forming two triangles $(1, 2, 3)$ and $(4, 5, 6)$. Because of the symmetry, the representation of each node should be identical throughout hypergraph message passing rounds. Hence, it is impossible for these models to conclude that $(1, 2, 3)$ is a triangle but $(4, 2, 3)$ is not, based only on the node representations, because they are identical. In contrast, NLMs with max arity 3 can solve them (as standard triangle detection problem in Table 1).

### A.5 The Time and Space Complexity of NLMs

Handling high-arity features and using deeper models usually increase the computational cost in terms of time and space. As an instance that use the architecture of RelNN, NLMs with depth $D$ and max arity $B$ takes $O(Dn^B)$ time when applying to graphs with size $n$. This is because both Expand and Reduce operation have linear time complexity with respect to the input size (which is $O(n^B)$ at each iteration). If we need to record the computational history (which is typically the case when training the network using back propagation), the space complexity is the same as the time complexity.

GNNs applied to $(B - 1)$-ary hyperedges and depth $D$ are equally expressive as RelNNs with depth $O(D)$ and max arity $B$. Though up to $(B-1)$-ary features are kept in their architecture, the broadcast message passing scheme scale up the complexity by a factor of $O(n)$, so they also have time and space complexity $O(Dn^B)$. Here the length of feature tensors $W$ is treated as a constant.

# B   Arity and Depth Hierarchy: Proofs and Analysis

## B.1   Proof of Theorem 3.1: Arity Hierarchy.

[11] have connected high-dimensional GNNs with high-dimensional WL tests. Specifically, they showed that the $B$-ary HO-GNNs are equally expressive as $B$-dimensional WL test on graph isomorphism test problem. In Theorem A.5 we proved that $B$-ary HO-GNNs are equivalent to NLM of maximum arity $B + 1$ in terms of expressiveness. Hence, NLM of maximum arity $B + 1$ can distinguish if two non-isomorphic graphs if and only if $B$-dimensional WL test can distinguish them.

However, Cai et al. [13] provided an construction that can generate a pair of non-isomorphic graphs for every $B$, which can not be distinguished by $(B - 1)$-dimensional WL test but can be distinguished by $B$-dimensional WL test. Let $G_B^1$ and $G_B^2$ be such a pair of graph.

Since NLM of maximum arity $B + 1$ is equally expressive as $B$-ary HO-GNNs, there must be such a NLM that classify $G_B^1$ and $G_B^2$ into different label. However, such NLM can not be realized by any NLM of maximum arity $B$ because they are proven to have identical outputs on $G_B^1$ and $G_B^2$.

In the other direction, NLMs of maximum arity $B + 1$ can directly realize NLMs of maximum arity $B$, which completes the proof.

## B.2   Upper Depth Bound for Unbounded-Precision NLM.

The idea for proving an upper bound on depth is to connect NLMs to WL-test, and use the $O(n^B)$ upper bound on number of iterations for $B$-dimensional test [34], and FOC formula is the key connection.

For any fixed $n$, $B$-dimensional WL test divide all graphs of size $n$, $\mathcal{G}_{=n}$, into a set of equivalence classes $\{\mathcal{C}_1, \mathcal{C}_2, \cdots, \mathcal{C}_m\}$, where two graphs belong to the same class if they can not be distinguished by the WL test. We have shown that NLMs of maximum arity $(B + 1)$ must have the same input for all graphs in the same equivalence class. Thus, any NLM of maximum arity $B + 1$ can be view as a labeling over $\mathcal{C}_1, \cdots, \mathcal{C}_m$.

Stated by Cai et al. [13], $B$-dimensional WL test are as powerful as $\text{FOC}_{B+1}$ in differentiating graphs graphs. Combined with the $O(n^B)$ upper bound of WL test iterations, for each $\mathcal{C}_i$, there must be an $\text{FOC}_{B+1}$ formula of quantifier depth $O(n^B)$ that exactly recognize $\mathcal{C}_i$ over $\mathcal{G}_{=n}$.

Finally, with unbounded precision, for any $f(n)$, NLM of maximum arity $B + 1$ and depth $f(n)$ can compute all $\text{FOC}_{B+1}$ formulas with quantifier depth $f(n)$. Note that there are finite number of such formula because the subscript of counting quantifiers is bounded by $n$.

For any graph in some class $\mathcal{C}_i$, the class can be determined by evaluating these FOC formulas, and then the label is determined. Therefore, any NLM of maximum arity $B + 1$ can be realized by a NLM of maximum arity $B + 1$ and depth $O(n^B)$.

## B.3   Graph Problems

| | Classification Tasks | Regression Tasks |
|---|---|---|
| $B = 4$ | 4-Clique Detection NLM[$O(1)$, 4] | 4-Clique Count NLM[$O(1)$, 4] |
| $B = 3$ | Triangle Detection NLM[$O(1)$,3]
Bipartiteness NLM[$O(\log n)$, 3]$^\star$
All-Pair Connectivity NLM[$O(\log n)$, 3]$^\star$
All-Pair Connectivity-$k$ NLM[$O(\log k)$, 3]$^\star$ | All-Pair Distance NLM[$O(\log n)$, 3]$^\star$ |
| $B = 2$ | $\text{FOC}_2$ Realization NLM[$\cdot$, 2] [14]
3/4-Link Detection NLM[$O(1)$, 2]
S-T Connectivity NLM[$O(n)$, 2]
S-T Connectivity-$k$ NLM[$O(k)$, 2] | S-T Distance NLM[$O(n)$, 2]
Max Degree NLM[$O(1)$, 2]
Max Flow NLM[$O(n^3)$, 2]$^\star$ |
| $B = 1$ | Node Color Majority: NLM[$O(1)$, 1] | Count Red Nodes: NLM[$O(1)$, 1] |

**Table 1:** The minimum depth and arity of NLMs for solving graph classification and regression tasks. The $^\star$ symbol indicates that these are conjectured lower bounds.

We list a number of examples for graph classification and regression tasks, and we provide the definitions and the current best known NLMs for learning these tasks from data. For some of the

problems, we will also show why they can not be solved by a simpler problems, or indicate them as open problems.

**Node Color Majority.** Each node is assigned a color $c \in \mathcal{C}$ where $\mathcal{C}$ is a finite set of all colors. The model needs to predict which color the most nodes have.

Using a single layer with sum aggregation, the model can count the number of nodes of color $c$ for each $c \in \mathcal{C}$ on its global representation.

**Count Red Nodes.** Each node is assigned a color of red or blue. The model needs to count the number of red nodes.

Similarly, using a single layer with sum aggregation, the model can count the number of red nodes on its global representation.

**3-Link Detection.** Given an unweighted, undirected graph, the model needs to detect whether there is a triple of nodes $(a, b, c)$ such that $a \neq c$ and $(a, b)$ and $(b, c)$ are edges.

This is equivalent to check whether there exists a node with degree at least 2. We can use a Reduction operation with sum aggregation to compute the degree for each node, and then use a Reduction operation with max aggregation to check whether the maximum degree of nodes is greater than or equal to 2.

Note that this can not be done with 1 layer, because the edge information is necessary for the problem, and they require at least 2 layers to be passed to the global representation.

**4-Link Detection.** Given an unweighted undirected graph, the model needs to detect whether there is a 4-tuple of nodes $(a, b, c, d)$ such that $a \neq c, b \neq d$ and $(a, b), (b, c), (c, d)$ are edges (note that a triangle is also a 4-link).

This problem is equivalent to check whether there is an edge between two nodes with degrees $\geq 2$. We can first reduce the edge information to compute the degree for each node, and then expand it back to 2-dimensional representations, so we can check for each edge if the degrees of its ends are $\geq 2$. Then the results are reduced to the global representation with existential quantifier (realized by max aggregation) in 2 layers.

**Triangle Detection.** Given a unweighted undirected graph, the model is asked to determine whether there is a triangle in the graph i.e. a tuple $(a, b, c)$ so that $(a, b), (b, c), (c, a)$ are all edges.

This problem can be solved by NLM [4,3]: we first expand the edge to 3-dimensional representations, and determine for each 3-tuple if they form a triangle. The results of 3-tuples require 3 layers to be passed to the global representation.

We can prove that Triangle Detection indeed requires breadth at least 3. Let $k$-regular graphs be graphs where each node has degree $k$. Consider two $k$-regular graphs both with $n$ nodes, so that exactly one of them contains a triangle[§]. However, NLMs of breadth 2 has been proven not to be stronger than WL test on distinguish graphs, and thus can not distinguish these two graphs (WL test can not distinguish any two $k$-regular graphs with equal size).

**4-Clique Detection and Counting.** Given an undirected graph, check existence of, or count the number of tuples $(a, b, c, d)$ so that there are edges between every pair of nodes in the tuple.

This problem can be easily solved by a NLM with breadth 4 that first expand the edge information to the 4-dimensional representations, and for each tuple determine whether its is a 4-clique. Then the information of all 4-tuples are reduced 4 times to the global representation (sum aggregation can be used for counting those).

Though we did not find explicit counter-example construction on detecting 4-cliques with NLMs of breadth 3, we suggest that this problem can not be solved with NLMs with 3 or lower breadth.

**Connectivity.** The connectivity problems are defined on unweighted undirected graphs. S-T connectivity problems provides two nodes $S$ and $T$ (labeled with specific colors), and the model needs to predict if they are connected by some edges. All pair connectivity problem require the model to

---

[§]Such construction is common. One example is $k = 2, n = 6$, and the graph may consist of two separated triangles or one hexagon

answer for every pair of nodes. Connectivity-$k$ problems have an additional requirement that the distance between the pair of nodes can not exceed $k$.

S-T connectivity-$k$ can be solved by a NLM of breadth 2 with $k$ iterations. Assume $S$ is colored with color $c$, at every iteration, every node with color $c$ will spread the color to its neighbors. Then, after $k$ iterations, it is sufficient to check whether $T$ has the color $c$.

With NLMs of breadth 3, we can use $O(\log k)$ matrix multiplications to solve connectivity-$k$ between every pair of nodes. Since the matrix multiplication can naturally be realized by NLMs of breadth 3 with two layers. All-pair connectivity problems can all be solved with $O(\log k)$ layers.

**Theorem B.1** (S-T connectivity-$k$ with NLM)**.** S-T connectivity-$k$ can not be solved by a NLM of maximum arity within $o(k)$ iterations.

*Proof.* We construct two graphs each has $2k$ nodes $u_1, \cdots, u_k, v_1, \cdots, v_k$. In both graph, there are edges $(u_i, u_{i+1})$ and $(v_i, v_{i+1})$ for $1 \le i \le k-1$ i.e. there are two links of length $k$. We then set $S = u_1, T = u_n$ and $S = u_1, T = v_n$ the the two graphs.

We will analysis GNNs as NLMs are proved to be equivalent to them by scaling the depth by a constant factor. Now consider the node refinement process where each node $x$ is refined by the multiset of labels of $x$'s neighbots and the multiiset of labels of $x$'s non-neighbors.

Let $C_j^{(i)}(x)$ be the label of $x$ in graph $j$ after $i$ iterations, at the beginning, WLOG, we have

$$C_1^{(0)}(u_1) = 1, C_1^{(0)}(u_n) = 2 C_1^{(0)}(u_1) = 1, C_1^{(0)}(v_n) = 2$$

and all other nodes are labeled as $0$.

Then we can prove by induction: after $i \le \frac{k}{2} - 1$ iterations, for $1 \le t \le i+1$ we have

$$C_1^{(u_t)} = C_2^{(i)}(u_t), C_1^{(v_t)} = C_2^{(i)}(v_t)$$

$$C_1^{(u_{k-t+1})} = C_2^{(i)}(v_{k-t+1}), C_1^{(v_{k-t+1})} = C_2^{(i)}(u_{k-t+1})$$

and for $i+2 \le t \le k-i-1$ we have

$$C_1^{(u_t)} = C_2^{(i)}(u_t), C_1^{(v_t)} = C_2^{(i)}(v_t)$$

This is true because before $\frac{k}{2}$ iterations are run, the multiset of all node labels are identical for the two graphs (say $S^{(i)}$). Hence each node $x$ is actually refined by its neighbors and $S^{(i)}$ where $S^{(i)}$ is the same for all nodes. Hence, before running $\frac{k}{2}$ iterations when the message between $S$ and $T$ finally meets in the first graph, GNN can not distinguish the two graphs, and thus can not solve the connectivity with distance $k-1$. $\qquad\square$

**Max Degree.** The max degree problem gives a graph and ask the model to output the maximum degree of its nodes.

Like we mentioned in 3-link detection, one layer for computing the degree for each node, and another layer for taking the max operation over nodes should be sufficient.

**Max Flow.** The Max Flow problem gives a directional graph with capacities on edges, and indicate two nodes $S$ and $T$. The models is then asked to compute the amount of max-flow from $S$ to $T$.

Notice that the Breadth First Search (BFS) component in Dinic's algorithm[35] can be implemented on NLMs as they does not require node identities (all new-visited nodes can augment to their non-visited neighbors in parallel). Since the BFS runs for $O(n)$ iteration, and the Dinic's algorithm runs BFS $O(n^2)$ times, the max-flow can be solved by NLMs with in $O(n^3)$ iterations.

**Distance.** Given a graph with weighted edges, compute the length of the shortest between specified node pair (S-T Distance) or all node pairs (All-pair Distance).

Similar to Connectivity problems, but Distance problems now additionally record the minimum distance from $S$ (for S-T) or between every node pairs (for All-pair), which can be updated using min operator (using Min-plus matrix multiplication for All-pair case).

# C Experiments

We now study how our theoretical results on model expressiveness and learning apply to relational neural networks trained with gradient descent on practically meaningful problems. We begin by describing two synthetic benchmarks: graph substructure detection and relational reasoning.

In the graph substructure detection dataset, there are several tasks of predicting whether there input graph contain a sub-graph with specific structure. The tasks are: *3-link* (length-3 path), *4-link*, *triangle*, and *4-clique*. These are important graph properties with many potential applications.

The relational reasoning dataset is composed of two family-relationship prediction tasks and two connectivity-prediction tasks. They are all *binary edge classification* tasks. In the family-relationship prediction task, the input contains the *mother* and *father* relationships, and the task is to predict the *grandparent* and *uncle* relationships between all pairs of entities. In the connectivity-prediction tasks, the input is the edges in an undirected graph and the task is to predict, for all pairs of nodes, whether they are connected with a path of length $\leq 4$ (*connectivity-4*) and whether they are connected with a path of arbitrary length (*connectivity*). The data generation for all datasets is included in Appendix C.

## C.1 Experiment Setup

For all problems, we have 800 training samples, 100 validation samples, and 300 test samples for each different $n$ we are testing the models on.

We then provide the details on how we synthesize the data. For most of the problems, we generate the graph by randomly selecting from all potential edges i.e. the Erdős–Rényi model. We sample the number of edges around $n, 2n, n \log n$ and $n^2/2$. For all problems, with $50\%$ probability the graph will first be divided into $2, 3, 4$ or $5$ parts with equal number of components, where we use the first generated component to fill the edges for rest of the components. Some random edges are added afterwards. This make the data contain more isomorphic sub-graphs, which we found challenging empirically.

**Substructure Detection.** To generate a graph that does not contain a certain substructure, we randomly add edges when reaching a maximal graph not containing the substructure or reaching the edge limit. For generating a graph that does contain the certain substructure, we first generate one that does not contain, and then randomly replace present edges with missing edges until we detect the substructure in the graph. This aim to change the label from "No" to "Yes" while minimizing the change to the overall graph properties, and we found that data generated using edge replacing is much more difficult for neural networks compared to random generated graphs from scratch.

**Family Tree.** We generate the family trees using the algorithm modified from [5]. We add people to the family one by one. When a person is added, with probability $p$ we will try to find a single woman and a single man, get them married and let the new children be their child, and otherwise the new person is introduced as a non-related person. Every new person is marked as single and set the gender with a coin flip.

We adjust $p$ based on the ratio of single population: $p = 0.7$ when more than $40\%$ of the population are single, and $p = 0.3$ when less than $20\%$ of the population are single, and $p = 0.5$ otherwise.

**Connectivity.** For connectivity problems, we use the similar generation method as the substructure detection. We sample the query pairs so that the labels are balanced.

## C.2 Model Implementation Details

For all models, we use a hidden dimension $128$ except for 3-dimensional HO-GNN and 4-dimensional NLM where we use hidden dimension $64$.

All model have $4$ layers that each has its own parameters, except for connectivity where we use the recurrent models that apply the second layer $k$ times, where $k$ is sampled from integers in $[2 \log n, 3 \log n]$. The depths are proven to be sufficient for solving these problems (unless the model itself can not solve).

All models are trained for 100 epochs using adam optimizer with learning rate $3 \times 10^{-4}$ decaying at epoch 50 and 80.

| Model | Agg. | 3-link $n=10$ | 3-link $n=30$ | 4-link $n=10$ | 4-link $n=30$ | triangle $n=10$ | triangle $n=30$ | 4-clique $n=10$ | 4-clique $n=30$ |
|---|---|---|---|---|---|---|---|---|---|
| 1-ary GNN | Max | $70.0_{\pm0.0}$ | $82.7_{\pm0.0}$ | $92.0_{\pm0.0}$ | $91.7_{\pm0.0}$ | $73.7_{\pm3.2}$ | $50.2_{\pm1.8}$ | $55.3_{\pm4.0}$ | $46.2_{\pm1.3}$ |
| | Sum | $100.0_{\pm0.0}$ | $89.4_{\pm0.4}$ | $100.0_{\pm0.0}$ | $86.1_{\pm1.2}$ | $77.7_{\pm8.5}$ | $48.6_{\pm1.6}$ | $53.7_{\pm0.6}$ | $55.2_{\pm0.8}$ |
| 2-ary NLM | Max | $65.3_{\pm0.6}$ | $54.0_{\pm0.6}$ | $93.0_{\pm0.0}$ | $95.7_{\pm0.0}$ | $51.0_{\pm1.7}$ | $49.2_{\pm0.4}$ | $55.0_{\pm0.0}$ | $45.7_{\pm0.0}$ |
| | Sum | $100.0_{\pm0.0}$ | $88.3_{\pm0.0}$ | $100.0_{\pm0.0}$ | $67.4_{\pm16.4}$ | $82.0_{\pm2.6}$ | $48.3_{\pm0.0}$ | $53.0_{\pm0.0}$ | $54.4_{\pm1.5}$ |
| 2-ary GNN | Max | $78.7_{\pm0.6}$ | $76.0_{\pm17.3}$ | $97.7_{\pm4.0}$ | $98.6_{\pm2.5}$ | $100.0_{\pm0.0}$ | $100.0_{\pm0.0}$ | $55.0_{\pm0.0}$ | $45.7_{\pm0.0}$ |
| | Sum | $100.0_{\pm0.0}$ | $51.2_{\pm7.9}$ | $100.0_{\pm0.0}$ | $45.7_{\pm7.6}$ | $100.0_{\pm0.0}$ | $49.2_{\pm1.0}$ | $61.0_{\pm5.6}$ | $54.3_{\pm0.0}$ |
| 3-ary NLM | Max | $100.0_{\pm0.0}$ | $100.0_{\pm0.0}$ | $100.0_{\pm0.0}$ | $100.0_{\pm0.0}$ | $100.0_{\pm0.0}$ | $100.0_{\pm0.0}$ | $59.0_{\pm6.9}$ | $45.9_{\pm0.4}$ |
| | Sum | $100.0_{\pm0.0}$ | $87.6_{\pm11.0}$ | $100.0_{\pm0.0}$ | $65.4_{\pm14.3}$ | $100.0_{\pm0.0}$ | $80.6_{\pm8.8}$ | $73.7_{\pm13.8}$ | $53.3_{\pm8.8}$ |
| 3-ary GNN | Max | $79.0_{\pm0.0}$ | $86.0_{\pm0.0}$ | $100.0_{\pm0.0}$ | $100.0_{\pm0.0}$ | $100.0_{\pm0.0}$ | $100.0_{\pm0.0}$ | $84.0_{\pm0.0}$ | $93.3_{\pm0.0}$ |
| | Sum | $100.0_{\pm0.0}$ | $84.1_{\pm18.6}$ | $100.0_{\pm0.0}$ | $61.1_{\pm15.0}$ | $100.0_{\pm0.0}$ | $95.1_{\pm7.3}$ | $80.5_{\pm0.7}$ | $66.2_{\pm19.6}$ |
| 4-ary NLM | Max | $100.0_{\pm0.0}$ | $100.0_{\pm0.0}$ | $100.0_{\pm0.0}$ | $100.0_{\pm0.0}$ | $100.0_{\pm0.0}$ | $100.0_{\pm0.0}$ | $82.0_{\pm1.7}$ | $93.1_{\pm0.2}$ |
| | Sum | $100.0_{\pm0.0}$ | $59.1_{\pm5.3}$ | $100.0_{\pm0.0}$ | $67.7_{\pm24.1}$ | $100.0_{\pm0.0}$ | $82.1_{\pm12.8}$ | $84.0_{\pm0.0}$ | $67.0_{\pm18.9}$ |

**Table 2:** Overall accuracy on relational reasoning problems. All models are trained on $n=10$, and tested on $n=30$. The standard error of all values are computed based on three random seeds.

| Model | Agg. | grand parent $n=20$ | grand parent $n=80$ | uncle $n=20$ | uncle $n=80$ | connectivity-4$^\P$ $n=10$ | connectivity-4$^\P$ $n=80$ | connectivity $n=10$ | connectivity $n=80$ |
|---|---|---|---|---|---|---|---|---|---|
| 1-ary GNN | Max | $84.0_{\pm0.3}$ | $64.8_{\pm0.0}$ | $93.6_{\pm0.3}$ | $66.1_{\pm0.0}$ | $72.6_{\pm3.6}$ | $67.5_{\pm0.5}$ | $85.6_{\pm0.3}$ | $75.1_{\pm1.9}$ |
| | Sum | $84.7_{\pm0.1}$ | $64.4_{\pm0.0}$ | $94.3_{\pm0.2}$ | $66.2_{\pm0.0}$ | $79.6_{\pm0.1}$ | $68.3_{\pm0.1}$ | $87.1_{\pm0.3}$ | $75.0_{\pm0.2}$ |
| 2-ary NLM | Max | $82.3_{\pm0.5}$ | $65.6_{\pm0.1}$ | $93.1_{\pm0.0}$ | $66.6_{\pm0.0}$ | $91.2_{\pm0.2}$ | $51.0_{\pm0.6}$ | $88.9_{\pm2.6}$ | $67.1_{\pm4.8}$ |
| | Sum | $82.9_{\pm0.1}$ | $64.6_{\pm0.1}$ | $93.4_{\pm0.0}$ | $66.7_{\pm0.2}$ | $96.0_{\pm0.4}$ | $68.3_{\pm0.5}$ | $84.0_{\pm0.0}$ | $71.9_{\pm0.0}$ |
| 2-ary GNN | Max | $100.0_{\pm0.0}$ | $100.0_{\pm0.0}$ | $100.0_{\pm0.0}$ | $100.0_{\pm0.0}$ | $100.0_{\pm0.0}$ | $100.0_{\pm0.0}$ | $84.0_{\pm0.0}$ | $71.9_{\pm0.0}$ |
| | Sum | $100.0_{\pm0.0}$ | $35.7_{\pm0.0}$ | $100.0_{\pm0.0}$ | $33.9_{\pm0.0}$ | $100.0_{\pm0.0}$ | $51.3_{\pm5.3}$ | $84.0_{\pm0.0}$ | $71.9_{\pm0.0}$ |
| 3-ary NLM | Max | $100.0_{\pm0.0}$ | $100.0_{\pm0.0}$ | $100.0_{\pm0.0}$ | $100.0_{\pm0.0}$ | $100.0_{\pm0.0}$ | $100.0_{\pm0.0}$ | $100.0_{\pm0.0}$ | $100.0_{\pm0.1}$ |
| | Sum | $100.0_{\pm0.0}$ | $35.7_{\pm0.0}$ | $100.0_{\pm0.0}$ | $50.8_{\pm29.4}$ | $100.0_{\pm0.0}$ | $77.8_{\pm11.8}$ | $100.0_{\pm0.0}$ | $88.2_{\pm8.0}$ |
| 3-ary NLM$_{HE}$ | Max | $100.0_{\pm0.0}$ | $100.0_{\pm0.0}$ | $100.0_{\pm0.0}$ | $100.0_{\pm0.0}$ | N/A | N/A | N/A | N/A |
| | Sum | $100.0_{\pm0.0}$ | $35.7_{\pm0.0}$ | $100.0_{\pm0.0}$ | $33.8_{\pm29.4}$ | N/A | N/A | N/A | N/A |

**Table 3:** Overall accuracy on relational reasoning problems. Models for family-relationship prediction are trained on $n=20$, while models for connectivity problems are trained on $n=10$. All model are tested on $n=80$. The standard error of all values are computed based on three random seeds. The 3-ary NLMs marked with "HE" have hyperedges in inputs, where each family is represented by a 3-ary hyperedge instead of two parent-child edges, and the results are similar to binary edges.

We have varied the depth, the hidden dimension, and the activation function of different models. We select sufficient hidden dimension and depth for every model and problem (i.e., we stop when increasing depth or hidden dimension doesn't increase the accuracy). We tried linear, ReLU, and Sigmoid activation functions, and ReLU performed the best overall combinations of models and tasks.

### C.3   Results

Our main results on all datasets are shown in Table 2 and Table 3. We empirically compare relational neural networks with different maximum arity $B$, different model architecture (GNN and NLM), and different aggregation functions (max and sum). All models use sigmoidal activation for all MLPs. For each task on both datasets we train on a set of small graphs ($n=10$) and test the trained model on both small graphs and large graphs ($n=10$ and $n=30$). We summarize the findings below.

**Expressiveness.** We have seen a theoretical equal expressiveness between GNNs and NLMs applied to hypergraphs. That is, a GNN applied to $B$-ary hyperedges is equivalent to a $(B+1)$-ary NLM. Table 2 and 3 further suggest their similar performance on tasks when trained with gradient descent.

Formally, triangle detection requires NLMs with at least $B=3$ to solve. Thus, we see that all NLMs with arity $B=2$ fail on this task, but models with $B=3$ perform well. Formally, 4-clique is realizable by NLMs with maximum arity $B=4$, but we failed to reliably train models to reach perfect accuracy on this problem. It is not yet clear what the cause of this behavior is.

**Structural generalization.** We discussed the structural generalization properties of NLMs in Section 4, in a learning setting based on fixed-precision networks and enumerative training. This setting can be *approximated* by training NLMs with max aggregation and sigmoidal activation on sufficient data.

We run a case study on the problem connectivity-4 about how the generalization performance changes when the test graph size gradually becomes larger. Figure 2 show how these models generalize to gradually larger graphs with size increasing from 10 to 80. From the curves we can see that only models with sufficient expressiveness can get 100% accuracy on the same size graphs, and among them the models using max aggregation generalize to larger graphs with no performance drop. 2-ary GNN and 3-ary NLM that use max aggregation have sufficient expressiveness and better generalization property. They achieve 100% accuracy on the original graph size and generalize perfectly to larger graphs.

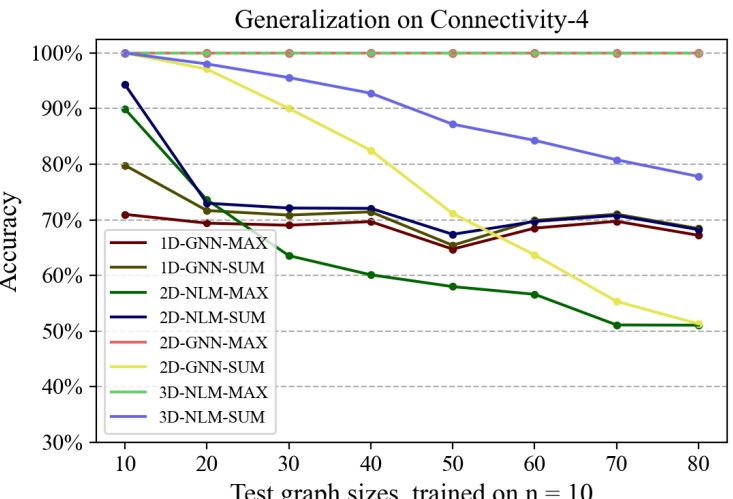

**Figure 2:** How the performance of models drop when generalizing to larger graphs on the problem connectivity-4 (trained on graphs with size 10).

