# OpenReview forum: "On the Expressiveness and Generalization of Hypergraph Neural Networks"
_logconference.io/LOG/2022/Conference — LoG 2022 Poster_

### Official Review · Reviewer_gMAT · 2022-10-12

**Overall Score:** 6
**Confidence:** 3

**Review:**

**Extended Abstract Summary:**

The authors propose a generalization of logical expressiveness studies done on message passing graph neural networks to message passing hypergraph neural networks. As one of the main results of the paper, akin to [1], they show that k-ary hypergraph neural networks are capable of realizing FOC-k logic which also relies on the logical equivalence of B-ary Higher order GNNs and (B+1) arity Neural Logic Machines (above a certain depth). The authors then present a generalization study, where they show that, when certain conditions hold (i.e. the availability of fixed precision aggregation function), a hypergraph neural network trained on small hypergraphs can generalize to hypergraphs of any size - when the arity is the same.

**Strong Points:**
1. The paper studies an important problem - learning on hypergraphs - given that most real world data are not limited to interactions of more than 2 objects at a given point.
2. The paper is well written and easy to understand and the proof sketches provide a good overview to the reader.
3. Experimental settings and results corroborate with the proposed theory


**Weak Points:**
1. A fixed precision aggregation function is impossible to realize in most ML settings and the authors must make it clear to the reader.
2. Scalability and data availability of the proposed learning framework is still a concern.
3. in the introduction, the authors claim that "Even when the inputs and outputs of models have only unary and binary relations, allowing intermediate hyperedge representations increases the expressiveness." - currently this is shown only empirically, and partially appears to contradict the necessary and sufficiency conditions presented.

**Initial Recommendation:**

Accept


**References:**

[1]. Pablo Barceló, Egor V Kostylev, Mikael Monet, Jorge Pérez, Juan Reutter, and Juan Pablo Silva. The logical expressiveness of graph neural networks. In ICLR, 2020. 3, 7, 13

---

### Official Review · Reviewer_XL5k · 2022-10-21

**Overall Score:** 6
**Confidence:** 3

**Review:**

# Summary

The paper investigates the theoretical properties of Hypergraph GNNs from two perspectives:
1. Expressive power with respect to the maximum k-ary of the hyperedges and the depth of the network.
2. Generalisation to graphs of arbitrary size from training on a dataset of finite size.

# Strenghts
- While the expressive power of hypergraph GNNs with respect to the order of the hyperedges has been studied before, less emphasis was placed on the number of layers. Therefore, I believe the paper makes a valuable contribution in this direction. In particular, Theorem 3.4 nicely formalises the intuition that lower k-ary requires more layers to solve a problem, whereas higher k-ary requires fewer layers. This is important to understand the effects that more expressive architectures have in function space.
- Besides expressivity, inductive generalisation of GNNs is another important problem that the paper tackles. While the assumptions of the result are heavily constraining (i.e. it requires the enumeration of all possible inputs of some zie), it does prove the generalisation to bigger graphs from a finite dataset is in principle possible.
- I appreciate the proof sketches supplied for each Theorem and they generally do a good job at conveying the reasoning behind the result.

# Weaknesses

- I think the relation to previous work should be improved and what is currently known should be more clearly highlighted. For instance, the paper says that one of its main takeaways is that *"Even when the inputs and outputs of models have only unary and binary relations, allowing intermediate hyperedge representations increases the expressiveness."*. However, this is a well-known fact because most of the theoretical analysis in this space assumes a graph input (i.e. only binary relations) followed by some "lifting procedure" that adds intermediate hyper-edges and optionally followed by a final projection. For instance, see the end of Section 2.2 in https://openreview.net/forum?id=lxHgXYN4bwl or Definition 8 in https://arxiv.org/abs/2106.12575.
- Fixed precision and unbounded precision models are not defined, but the notion is central in all the results. There is a definition of *fixed-precision aggregation function* that comes towards the end of the paper and I am not sure what its precise relationship to *fixed-precision models* is.
- I believe Theorems 3.1 and 3.2  should simply be a corollary as everything follows from the equivalence between HO-GNNs and NLMs and known results about HO-GNNs. Since HO-GNNs follow a strict hierarchy and NLM[B + 1] are equivalent to HO-GNNs[B], Theorem 3.1 follows. Similarly, given the connections between HO-GNNs -> WL -> FOC_k, the lower and upper bounds in Theorem 3.2 easily follow.

# Further Suggestions
- The generalisation of GNNs to bigger graphs is also analysed in https://arxiv.org/abs/2207.07888 (NeurIPS 2022) from a more empirical perspective, but under more realistic assumptions. I believe this is relevant for Section 4 and should be cited. (Note: I am NOT an author of this paper)

---

### Official Review · Reviewer_YGPC · 2022-10-26

**Overall Score:** 6
**Confidence:** 3

**Review:**


In summary, this paper shows the expressiveness of NLM with different depths and arities, and it characterizes the generalization power of the trained NNs.

Strength:
This paper provides the arity hierarchy and depth hierarchy to verify the influence of these two aspects on model expressiveness.

Weakness:
The motivation for the depth hierarchy is not clear. In general, GNNs are usually shallow, i.e., two or three layers, and will not increase the model depth along with the node numbers, e.g., three-layer GNNS Ho-GNN as mentioned in [9]. The statements in lines 117 and 118 need some references.

---

### Official Review · Reviewer_1xZj · 2022-10-26

**Overall Score:** 6
**Confidence:** 2

**Review:**

### Summary:
The paper theoretically analyzes the expressiveness and generalization capabilities of neural networks operating on hypergraphs. To this end, the authors first prove that higher-order GNNs (HO-GNNs) and Neural Logic Machines (NLMs) can realize the same functions, i.e., they are equivalent in terms of expressiveness. They subsequently show hierarchical expressivity relations depending on the size of hyper-edges and depth of the neural networks for the class of NLM models. Furthermore, a theorem on the generalization of these models from small training graphs to larger testing graphs is presented.

### Reasons for score:
Overall, I recommend this submission to be accepted. The results presented in the paper seem interesting and relevant to me. The paper is well structured and does a good job of introducing important concepts and definitions despite the short 4 page format.
I would like to emphasize, however, that I do not work in this field and cannot confidently judge the novelty and correctness of proofs.

### Pros:
1. Better understanding the expressiveness of hypergraph neural networks and graph neural networks in general is an important direction for the GNN community. For higher-order GNNs, in particular, a well-grounded understanding of their capabilities is needed to justify the increased computational cost.
2. To me, theorem 2.1 (equivalence of HO-GNNs and NLMs in terms of expressiveness) is interesting as it allows the authors and future work to analyze the two families of neural networks in a joint framework.
3. The discussion on the expressive power of the graph neural networks with varying depths and edge arities might have practical implications for the design of such networks.
4. Proofs and proof sketches are provided for all theorems. Most claims are also supported by a small empirical study in the appendix.

### Cons:
1. Theorem 4.1 (generalization under complete presentation) is interesting, but I wonder how large the integer $N$ is going to be in practice? Therefore, it is unclear to me how relevant this theorem is in practical applications.
2. It could be stated more clearly how theorem 3.1 relates to (and extends) previous results for higher-order GNNs.
3. Since I am not familiar with the work on first-order logic inspired GNNs and NLMs, more pointers to relevant literature would be helpful for me to understand the connection with higher-order GNNs better.
4. “[…] allowing them to have a depth that is dependent on the number of nodes […] can substantially increase their expressive power” (line 117). I assume this has been discussed elsewhere already and might need a citation.

### Minor comments:
- The FOC acronym is used but not explained in the introduction
- The C extension of first-order logic (FOC) should be explained, ideally already in the introduction; I found a useful explanation in: *Grohe, Martin. "The logic of graph neural networks." (2021)*, for example.
- I would maybe change the wording in the introduction: “under certain ~~realistic~~ assumptions” (line 31) because the enumeration assumption in theorem 4.1 does not strike me as being particularly realistic.
- Some typos should be corrected (e.g. “m” in line 42)

### Questions:
- It is not entirely clear to me what part of the paper the sentence “Even when the inputs and outputs of models have only unary and binary relations, allowing intermediate hyperedge representations increases the expressiveness” (introduction, line 29) refers to – especially the part about input/output relations.

---

### Meta-Review · Area_Chair_KDEg · 2022-11-16

**Confidence:** 4
**Recommendation:** Borderline and needs further discussi…

**Meta Review:**

This paper examines the expressive power and generalization capabilities of two previously proposed neural models for hypergraphs. All reviews gave the paper a weak acceptance score and noted that the questions raised are interesting and important. A significant criticism was made regarding the fact that the authors failed to discuss related previous work - analyzing the expressive power of high-order GNNs in an area that is relatively well-researched. The authors added citations to two relevant works, but I believe the paper should be reorganized in order to explain the differences between the presented results and previous ones. Specifically, lower and upper bounds of high-order GNNs were previously described in terms of the WL hierarchy. The results on generalization to unseen sizes are interesting as well but again fail to discuss several previous works that discussed studied this problem in the past.

Considering the insufficient positioning of the paper in comparison with previous research, it is recommended that the paper be rejected in order for the authors to make the necessary changes. Furthermore, I suggest that, in the future, a longer version of the paper be submitted in order to include a detailed definition of the architectures and the relationship between this work and previous work.

---

### Decision · Program_Chairs · 2022-11-22

Accept (Poster)